# A New Era of Integration between Multiomics and Spatio-Temporal Analysis for the Translation of EMT towards Clinical Applications in Cancer

**DOI:** 10.3390/cells12232740

**Published:** 2023-11-30

**Authors:** Adilson Fonseca Teixeira, Siqi Wu, Rodney Luwor, Hong-Jian Zhu

**Affiliations:** 1Department of Surgery, The Royal Melbourne Hospital, The University of Melbourne, Parkville, VIC 3050, Australiawus8@student.unimelb.edu.au (S.W.); rluwor@unimelb.edu.au (R.L.); 2Huagene Institute, Kecheng Science and Technology Park, Pukou District, Nanjing 211800, China; 3Fiona Elsey Cancer Research Institute, Ballarat, VIC 3350, Australia; 4Health, Innovation and Transformation Centre, Federation University, Ballarat, VIC 3350, Australia

**Keywords:** cancer metastasis, EMT, genomics, multiomics, proteomics, secretome, transcriptomic

## Abstract

Epithelial-mesenchymal transition (EMT) is crucial to metastasis by increasing cancer cell migration and invasion. At the cellular level, EMT-related morphological and functional changes are well established. At the molecular level, critical signaling pathways able to drive EMT have been described. Yet, the translation of EMT into efficient diagnostic methods and anti-metastatic therapies is still missing. This highlights a gap in our understanding of the precise mechanisms governing EMT. Here, we discuss evidence suggesting that overcoming this limitation requires the integration of multiple omics, a hitherto neglected strategy in the EMT field. More specifically, this work summarizes results that were independently obtained through epigenomics/transcriptomics while comprehensively reviewing the achievements of proteomics in cancer research. Additionally, we prospect gains to be obtained by applying spatio-temporal multiomics in the investigation of EMT-driven metastasis. Along with the development of more sensitive technologies, the integration of currently available omics, and a look at dynamic alterations that regulate EMT at the subcellular level will lead to a deeper understanding of this process. Further, considering the significance of EMT to cancer progression, this integrative strategy may enable the development of new and improved biomarkers and therapeutics capable of increasing the survival and quality of life of cancer patients.

## 1. Introduction to EMT

Epithelial-mesenchymal transition (EMT) is recognized as a critical turning point for epithelial cells to acquire a mesenchymal phenotype with new biological functionalities. EMT is a process that happens to epithelial cells with appropriate external stimuli, where downregulation of epithelial traits and upregulation of mesenchymal characteristics allow these cells to escape mechanical and biochemical constraints while facilitating cell detachment and movement [1,2,3]. This process involves alterations in cell junctions, polarity, and cytoskeletal arrangements, which are all important factors in determining the cell phenotype [1,2,4]. Due to its essential roles in both physiological and disease states, EMT gained vast interest from researchers in different biomedical fields, especially in cancer research.

During cancer progression, particularly for carcinomas, EMT-related alterations enhance the migratory and invasive potential of malignant cells, critically contributing to the early stages of metastasis. In recent years, an enormous number of epigenomics, transcriptomics, and proteomics studies have helped to characterize the regulatory mechanisms and consequences of EMT on cancer [5,6,7,8]. However, a comprehensive view of the complex alterations that occur at multiple molecular levels in malignant cells undergoing EMT is still missing.

Here, the current understanding achieved by independently using different omics to depict the EMT is reviewed. Large efforts have been made to portray the changes associated with the epithelial and mesenchymal phenotypes of carcinoma cells. Still, limitations imposed by available methods and experimental design have delayed and compromised the precise characterization of EMT drivers and the impact of their crosstalk during cancer progression. Therefore, implementing an approach focused on spatio-temporal multiomics may be the next step still required for the translation of EMT-related knowledge into clinics. This strategy may allow the development of efficient methods to detect, prevent, and treat disease progression, thus, improving the outcome of cancer patients.

### 1.1. Molecular Regulation of EMT

EMT is a complex event that involves multiple levels of molecular regulation that dynamically change throughout the process. Several positive and negative molecular regulators have been associated with this phenotypic alteration while several others are yet to be elucidated. Considering the multiple biological processes associated with the EMT, characterizing its regulatory mechanisms served as the basis for several in vitro and in vivo studies and the development of useful techniques for the diagnosis and treatment of different diseases [9,10,11].

EMT begins with the activity of EMT-inducers in the tissue environment, such as Epithelial growth factor (EGF), Fibroblast growth factor (FGF) and Transforming growth factor-beta (TGF-β) [12,13,14,15,16,17]. Their interaction with corresponding receptors in the plasma membrane leads to the activation of ligand-receptor complexes and intracellular signal transduction, which consequently initiates the activation/suppression of downstream cellular effectors. Such intracellular effectors are tightly regulated by post-translational modifications critically impacting their stability, localization, and activity [18,19,20,21,22]. Additionally, alterations in intracellular effectors contribute to the expression of target genes associated with EMT, resulting in a feedback loop that could either increase the acquisition of mesenchymal characteristics or reverse this phenotypic transition. To promote EMT, these effectors downregulate epithelial proteins (e.g., E-cadherin and ZO-1) and upregulate mesenchymal proteins (e.g., N-cadherin, Vimentin), leading to an elongated phenotype and a front-rear polarity typical of cells undergoing EMT [23] (Figure 1). Importantly, as these changes (particularly those observed at the cellular level) are well established, their detection helps researchers to investigate further EMT-associated changes in vitro and in vivo.

As EMT may incur drastic disturbances to the tissue structure, this phenotypic transition is tightly regulated under physiological conditions to control the extent and duration of EMT [15,24]. Negative regulatory processes may include the suppression of EMT-promoting signaling pathways through the expression of inhibitory proteins, some of which are induced by these signaling pathways themselves as a negative feedback mechanism [25,26,27,28,29]. Otherwise, extracellular autocrine or paracrine signals inducing a mesenchymal-epithelial transition (MET) can also balance this process [24,30,31]. Restoring the levels of epithelial and mesenchymal proteins during MET facilitates the re-establishment of an epithelial cell phenotype [31,32].

As EMT and MET are critical for homeostasis, deregulated alterations in the mechanisms that control these processes unequivocally contribute to disease development and progression. Indeed, modifications driving EMT and MET in cancer are not easily controlled, being amplified by the dynamic modification of the tumor microenvironment and the crosstalk between cancer cells and non-cancer cells. Such deregulated changes critically correlate with cancer metastasis and resistance to therapy, which both are major causes of death for cancer patients. Therefore, characterizing the phenotypic alterations and the precise plethora of molecular mechanisms that collectively control EMT/MET in malignant cells may help to overcome existing limitations in the diagnosis and treatment of cancer patients.

### 1.2. EMT as an ‘Accomplice’ to Cancer Metastasis

Although triggered by common inducers and characterized by similar molecular and cellular changes, EMT occurs in three contextually different situations. According to the outcomes, these are classified as type I EMT—embryogenesis, type II EMT—wound healing or fibrosis, and type III EMT—cancer progression and metastasis [1]. As metastasis is a multistep process, cancer cells must undergo several steps of change to leave the primary tumor and colonize distant sites. To successfully migrate and invade surrounding tissues, cancer cells undergo a profound change in a profile of proteins controlling their interactions with neighboring cells, the basement membrane, and the extracellular matrix (ECM). After blood/lymph vessel intravasation, cancer cells disseminate, extravasate the vasculature and seed secondary sites [33,34]. Then, cancer cells are expected to reverse their phenotype by MET to form micrometastases that may later result in macroscopic lesions [32,35,36,37,38] (Figure 2).

Interestingly, cancer cells undergoing EMT may show heterogeneous degrees of epithelial and mesenchymal characteristics, including intermediate states known as ‘partial’ (p)EMT [35,36,39] (Figure 1). This is often reflected in the ratio of EMT markers in cancer cells, such as E-cadherin and N-cadherin. Whereas cancer cells that complete the EMT may benefit from single-cell migration, the epithelial traits conserved by cancer cells with pEMT may also enhance their dissemination in the circulation as a cluster, facilitating their colonization at distant organs [40,41,42,43,44]. The existence of a pEMT adds complexity and difficulties to omic-based studies. Averaged quantifications of gene/protein levels in multiple cancer cells cannot adequately represent the heterogeneous phenotypes of individual cells showing distinct EMT degrees in between themselves and over time.

Noteworthy, although many of the EMT outcomes are well described, further analyses of regulatory mechanisms are yet to be detailed. This need is associated with the complexity of this process and may be attributed to the large scale of molecular adaptations happening at different stages of EMT that are regulated by the crosstalk between multiple signaling pathways. Crucial molecular pathways and traditional transcription factors responsible for driving the molecular reprogramming that is associated with the EMT have been reasonably characterized so far, but understanding their crosstalk during EMT is still a complex task. An alternative to obtaining an unbiased portrait of changes during EMT is the simultaneous use of methods such as RNA and protein microarrays, RNA-sequencing (RNA-seq) and mass spectrometry (MS). Still, these technologies were rarely combined in EMT studies. Therefore, in the following sections, we review EMT-related changes established by studies at either DNA, RNA, or protein level, before discussing the integration of multiple platforms. As such, the limitations of the individual methods are minimized and compensated to better integrate them into mechanistic studies and clinical applications.

## 2. Benefits and Pitfalls of Analyzing Cancer Cell EMT at the DNA/RNA Level

In order to evaluate the gene expression profile of cancer cells undergoing EMT, it is important to successfully recapitulate this process under experimental conditions. Different models have been used to study EMT-related alterations and their impact on cancer progression. In vitro studies, as those discussed here, frequently rely on the treatment of cancer cells with known EMT inducers. Alternatively, the exogenously introduced overexpression of such molecules can also be used to investigate downstream consequences on cell transcriptomes. Moreover, while EMT is known to drive therapeutic resistance, recent studies have demonstrated that drug treatment can also induce EMT in cancer cells [45,46]. Noteworthy, because the phenotypic changes associated with EMT are very well-established, researchers can use different approaches to induce EMT that are easily verifiable, increasing the reliability of the conclusions obtained. In this section, selected studies are briefly discussed aiming at exemplifying the main findings in the field while allowing further comparison with changes observed at protein level in cancer cells undergoing EMT.

Among other results, evaluating gene expression alterations that are associated with cancer cell EMT contributed to portraying the many modifications triggered during cancer progression [47,48,49,50]. Such studies helped to determine not only the duration of EMT-related alterations at the molecular level but also to establish how different effectors cooperate to regulate EMT over time. In bladder carcinoma cells, for example, while alterations in typical EMT markers (e.g., Keratins and Vimentin) were only observed 24 h after FGF-1 treatment, changes in the expression of different transcription factors (e.g., *ETS* and *JUNB*) were reported to start and end shortly (2 h) after cell stimulation [47]. Interestingly, alterations in the gene expression of transcription factors were also observed in prostate cancer cells undergoing EMT [51]. Exogenous expression of Runx2 in these cancer cells led to *SOX9* and *SNAI2* upregulation, reinforcing the existence of cooperation between transcription factors notoriously associated with EMT towards the amplification of this process [51]. Indeed, the relevance of EMT-TFs for the regulation of the EMT program over time has been recently reinforced by Frey and colleagues investigating the pro-metastatic phenotype of SMAD4-defective colorectal cancer cells [52]. Although SMAD4 is a transcription factor critical to canonical TGF-β and Bone morphogenetic protein (BMP) signaling activity, overexpression of the EMT-TF SNAI1 in SMAD4-mutant colorectal cancer cells compensated for this deficiency, inducing cancer cell elongation and invasion in vitro [52]. Accordingly, SMAD4-mutant cells overexpressing SNAI1 also exhibited typical transcriptome changes expected in cells undergoing EMT, including the activation of other pro-EMT molecular pathways (e.g., Wnt signaling pathway) and increased activity of other EMT-TFs (e.g., FOXO4) [52]. These studies emphasize the complexity of the EMT program where distinct signaling pathways can cooperate to enhance this phenotypic shift or compensate each other in case of specific losses or deficiencies.

Interestingly, different studies have reported the failure to establish a good correlation between particular pairs of mRNA and proteins in the EMT context. Although protein degradation must certainly account for part of this observation, post-transcriptional modifications are also likely to contribute to this result. Post-transcriptional modifications respond to an additional layer of complexity in the regulatory mechanisms of the EMT by modifying mRNA stability. For instance, Shapiro and colleagues (2011) described little overlap between genes regulated at the expression level and at the splicing level in HMLE mammary cells undergoing EMT [53]. Yet, both regulatory mechanisms were reported to largely impact the levels of mRNA transcribed from genes associated with cytoskeleton assembling and cell-cell junctions [53]. The impact of splicing events on mRNA levels during EMT was also characterized in prostate cancer cells where 900 differential alternative splicing events were described, impacting the expression of several genes associated with EMT, migration and invasion [49]. Therefore, these and other studies in the field found a tight control of the EMT program that actively operates on RNA species and, more specifically, not only precedes but also follows transcription.

Still, whereas post-transcriptional modifications partially respond to changes in RNA levels, alterations in the epigenome are also expected to contribute to this process. For example, Taube and colleagues (2013) reported a 10-fold gain in the DNA methylation levels at the promoter of the microRNA-203 (miR-203) in Twist-overexpressing HMLE cells [54]. Interestingly, reduced levels of miR-203 were also observed in breast cancer cell lines naturally exhibiting mesenchymal traits (e.g., MDA-MB-231 breast cancer cells), whereas increased miR-203 levels were detected in epithelial counterparts (e.g., MCF-7 breast cancer cells) [54]. Also investigating epigenetic alterations, Peixoto et al. (2019) reported an association between EMT and increased methylation in histone H3 residues, such as in lysine 4 (H3K4) [55]. Moreover, 23% of all upregulated genes characterized in this study coincided with genes potentially activated by H3K4 bi-methylation (H3K4me2) [55]. In fact, increased levels of the inducer mark H3K4me2 and decreased levels of the repressive mark H3K27me3 were found at the promoter of Matrix metalloproteinase 9 (*MMP9*), a proteinase typically associated with ECM remodeling and cell invasion [55].

In addition to alterations previously described that can regulate epithelial and mesenchymal characteristics, the activity of non-coding RNA can also interfere with mRNA levels. Loboda et al. (2011), for example, reported a negative correlation between the expression of miR-200 family members and EMT-related genes in colorectal cancers (CRCs) [50]. In head and neck squamous cell carcinomas (HNSCCs), the expression of the long non-coding RNA (lncRNA) lnc-LCE5A-1 and lnc-KCTD6-3 was associated with reduced overall survival, while the ectopic expression of this lncRNA decreased Vimentin mRNA levels and impaired HNSCC cell migration in vitro [56]. Reinforcing the importance of characterizing alterations in non-coding RNA, Liao et al. (2017) showed that the impact of TGF-β treatment on the global levels of lncRNA is even higher than its impact on mRNA levels in MCF10A cells [57]. Moreover, the knockdown of the RP6-65G23.5 lncRNA in this model decreased E-cadherin and ZO-1 expression while increasing Vimetin, N-cadherin and Fibronectin levels, and inducing cell migration and invasion [57].

Beyond the evaluation of mRNA levels using in vitro models, transcriptome studies of human tumor samples have consolidated the association between cancer cell EMT and poor prognosis, highlighting the significance of transcriptome analysis in clinical applications. For example, Marquardt et al. (2014) reported the enrichment of an EMT signature in advanced hepatocellular carcinoma (HCC) samples which was also correlated with reduced survival and increased recurrence [58]. Interestingly, several genes differentially regulated in this molecular signature are known targets of TGF-β, Notch, and Vascular endothelial growth factor (VEGF), and are related to cell adhesion, cytoskeleton remodeling and EMT [58]. Nevertheless, it is well-known that changes in EMT-related genes may be impacted by the occurrence of non-cancer cells within tumor masses. This is particularly concerning when considering stroma-rich tumors, where cells from mesenchymal origin expressing high levels of mesenchymal-related genes may mislead the interpretation of bulk transcriptomic analyses of tumor samples with low cancer cell purity [59,60,61]. To overcome this limitation, the increasing use of single-cell (sc)RNA-seq in cancer studies has helped to better understand the significance of such modifications [62,63]. Instead of necessarily refuting previous studies, new methodologies may point to directions otherwise counterintuitive [60,64]. For instance, while cancer cell EMT and invasion are commonly associated, the invasive side of endometrioid adenocarcinomas is reported to be poorly represented by cancer cells expressing EMT-related genes. Such an EMT-related pattern, however, is observed in the endometrial side of these tumors [64]. Moreover, whereas the expression of epithelial and mesenchymal traits are often anticorrelated, increasing evidence highlights the coexistence of such phenotypes. Indeed, scRNA-seq coupled with trajectory inference analysis of HNSCCs has demonstrated that while some cancer cells show low levels of epithelial and mesenchymal markers, they can transition into a phenotype that simultaneously expresses high levels of epithelial- and mesenchymal-related genes [65]. Additionally, other HNSCC cancer cells can sustain elevated epithelial traits while dynamically regulating the expression of mesenchymal genes [65]. Noteworthy, these and other observations have driven the development of methods that enable estimating the composition of tumor samples analyzed by bulk transcriptomics through a combination of deconvolution and inference of tumor purity [66,67,68,69]. These approaches improve our comprehension of the relevance of cancer cells and stromal components during cancer progression while additionally increasing the accuracy of analyses relying on EMT signatures.

As discussed in this section, very important findings regarding cancer cell EMT originated from studies exploring alterations at the DNA/RNA level. Methodologies such as microarrays, RNA-seq, scRNA-seq, and assay for transposase-accessible chromatin with sequencing (ATAC)-seq can be used in a less biased way to reveal crucial players that must then be further investigated in more restricted conditions using additional controls. Still, however important, RNA shows limited functions and the main phenotypic changes observed during EMT come from protein activity. Thus, using similar EMT models to investigate the proteome of cancer cells is also critical to deepening our understanding of the main drivers involved in this phenotypic transition.

## 3. Proteomics Translated from Bench-to-Bedside

Characterizing specific protein functions has always been greatly important in cancer research due to the multiple roles these molecules can play in distinct cellular processes. Changes in protein level, localization, and activity—which are regulated by post-translational modifications and interaction with other molecules—largely affect EMT. However, similar to studying alterations in mRNA levels, focusing on one or a few proteins may be detrimental to portraying the complex scenario that involves EMT. Proteome studies help to overcome this limitation by reducing bias while analyzing this process. Moreover, if integrated into the transcriptome and epigenome analysis, proteomics can be used to identify crucial biomarkers and molecular pathways not only altered as a consequence of EMT but also responsible for driving this molecular program.

In this section, we will discuss how methodologies applied to describe the proteome of cancer cells undergoing EMT established the fundamental concepts associated with cancer cell migration and invasion (Section 3.1). Techniques such as MS and tissue microarray (TMA) have been used in different models, allowing not only the identification of EMT-related proteins, but also changes in their stability, activity, and localization in response to post-translational modifications (Figure 3). Because resistance to therapy and metastasis are closely associated with EMT, we next focus on studies that characterized the proteome of human tumor samples (Section 3.2) and biological fluids from cancer patients (Section 3.3). Understanding past studies that explored EMT-related biomarkers in vitro and in clinical samples may improve the diagnosis and treatment of cancer patients.

### 3.1. Using In Vitro Models to Analyze the Proteome of Cancer Cells Undergoing EMT

As previously discussed regarding the characterization of cancer cell epigenome or transcriptome, the use of cell cultures as models helps us to understand how individual or combined signaling pathways are altered and/or alter cancer cell EMT. Again, strategies employed for this purpose mainly include the induction of cancer cell EMT by stimulation with EMT-related growth factors or ectopic expression of EMT-inducers.

For example, TGF-β treatment was shown to induce EMT by triggering global alterations in the expression of several cytoskeleton proteins, such as β-actin and Cofilin1 [70], but also by indirectly changing the localization of cell adhesion proteins (e.g., ZO-1) through ADAM metallopeptidase domain 12 (ADAM12L) upregulation [71,72]. Similarly, downregulation of proteins associated with cell adhesion (e.g., E-cadherin and ZO-1) was reported in gastric cancer (GC) cells overexpressing Ubiquitin like with PHD and ring finger domains 2 (UHRF2) [73]. In this context, UHRF2 reduces E-cadherin mRNA levels by binding to the E-cadherin promoter and repressing the transcription of this epithelial marker [73]. UHRF2 overexpression was also associated with the upregulated activity of typical EMT-transcription factors (EMT-TFs), and the interaction between UHRF2 and Transcription factor 7 like 2 (TCF7L2) was further demonstrated in this model [73]. In another example, overexpression of Mitogen-activated protein kinase kinase (MEK)5 overexpression in breast cancer cells (BCs) resistant to Tumor necrosis factor-alpha (TNF-α) upregulated Vimentin and downregulated Keratins 8 and 19 [74]. Accordingly, MEK5 overexpression also induced EMT-related morphology changes and colony formation [74].

It is interesting to note that many studies evaluating alterations in the proteome of cells undergoing EMT frequently report cytoskeleton and other structural proteins among the main molecules differentially expressed. As discussed before, while a proteome-based investigation enables the broadest identification of global changes, the sensitivity of most methods is still low if compared with techniques used to evaluate modifications at the DNA/RNA level. Therefore, improving the sensitivity of these methods is a notorious goal for most studies that attempt to reveal the proteome of cancer cells or patient samples, and the analysis of sub-proteomes can help to achieve this aim.

#### 3.1.1. Compartmentalization and Specificity of Sub-Proteome

Although analyzing whole cell lysates can lead to a wide view of the changes that cancer cells are subjected to when transitioning towards a mesenchymal phenotype, many important mediators can be expressed at low levels. Therefore, the relevance of these less abundant proteins can be masked by highly expressed proteins. As an alternative to solve this problem, the fractionation of whole cells before protein extraction can enrich protein samples and allow the identification of otherwise overlooked EMT-related proteins. While improving the detection of poorly expressed proteins, the analysis of sub-proteomes also allows researchers to visualize the distribution of these molecules across distinct cell compartments such as the nucleus, cytoplasm, and membranes.

Chen et al. (2011), for example, used this approach to study the membrane proteome of 21D1 cells (HRAS-transformed MDCK cell clones) undergoing TGF-β-induced EMT [75]. In addition to a shift towards a pattern of integrin-mediated adhesion, researchers showed that Wnt5a was the top upregulated protein in this model and that its expression repressed the canonical Wnt signaling pathway in 21D1 cells [75]. Interestingly, a similar elevation in integrin membrane levels has also been described in EMT models of ovarian and breast cancers [76,77]. Sub-proteome analysis of ovarian adenocarcinoma cells demonstrated that EGF treatment upregulates Integrin α2 in the membrane fraction of Caov-3 cells during EMT [76]. In BC cells, Palma and collaborators (2016) demonstrated that SNAIL overexpression drives cell elongation and increases Integrin β1 expression in the membrane fraction of MCF7 cells [77]. Also evaluating the sub-proteome of breast cells, Silvestrini and colleagues (2020) described the upregulation of Ubiquitin specific peptidase (USP)47 in the cytoplasm fraction of TGF-β2-treated MCF10A cells [78]. Interestingly, treatment with a pharmacological inhibitor targeting USP47 (P5091) upregulated E-cadherin and downregulated SNAIL both in the nucleus and cytoplasm, while reducing morphological changes otherwise observed in MCF10A cells treated with TGF-β2 [78].

As observed by the results discussed here, cell fractionation is indeed a potent strategy to unmask changes in the levels of poorly expressed proteins. Still, many important proteins associated with the EMT program do not remain inside the cell. For example, proteins associated with cell-cell communication and ECM structural components have their main activity in the extracellular space. Thus, enrichment methods should be expanded to further investigate alterations in the secretome that are associated with EMT.

#### 3.1.2. What Is on the Outside Matters: Secretome and Cell Communication

The study of sub-proteomes with a focus on secreted molecules contributes to the characterization of proteins that are critically associated with EMT by regulating, for instance, cell-cell communication and ECM remodeling. Traditionally, the global evaluation of these proteins has been referenced as *secretome*. Still, different routes of secretion exist, and some studies have also investigated non-canonical pathways driving protein secretion through extracellular vesicles (EVs). Although many types of EVs exist and differ in biogenesis and composition, most results converge towards a critical role for EVs in mediating cell-cell communication, as comprehensively reviewed by Greening [79], Xu [80], Raposo [81], Couch [82], and respective collaborators. Many studies have already demonstrated the contribution of EVs to different processes associated with cancer growth and progression. Here, studies investigating the profile of proteins secreted by cancer cells (mediated or not by EVs) are discussed with a focus on their relevance to EMT. Importantly, different EV populations were historically described as *exosomes* only considering their small diameter (30–100 nm) and without further characterization of their biogenesis. This nomenclature is no longer recommended in most circumstances [83], but the term will be used henceforward in this review respecting original publications.

Large alterations in the secretome of cells undergoing EMT have been reported. For instance, 287 differentially expressed proteins were described in the secretome of 21D1 cells compared with the secretome of parental MDCK cells, many associated with ECM remodeling [84,85]. Interestingly, 71% of these differentially expressed proteins exhibited concordant altered mRNA levels, including Collagen α-2I and Serine peptidase inhibitor Kazal type (SPINK)5, suggesting a major impact on regulatory mechanisms acting at the DNA/RNA level [84,85]. In BC cells, HMGA1 knockdown induced a cobblestone-like morphology in MDA-MB-231 cells [86], while reducing the secretion of several proteins associated with decreased cancer relapse and distant metastasis-free survival [e.g., Serpin E1 (SERPINE1) and Plasminogen activator, urokinase (PLAU)] [87]. Also, using BC models, Erin et al. (2018) compared alterations in the proteome of poorly metastatic 67NR cells and highly metastatic 4T1 cells [88]. Strikingly, whereas several proteins associated with ECM composition and remodeling were similarly expressed in whole cell lysates of both cell lines, proteins such as BMP1, Fibulin-4, MMP-3, and MMP-9 were increased in the secretome of metastatic cells [88].

Focused on specific alterations that impact non-canonical routes of secretion, Tauro and collaborators (2013) compared the proteome of exosomes secreted by epithelial MDCK cells and mesenchymal 21D1 clones to establish the contribution of these EVs to EMT [89]. Several proteins downregulated in 21D1-exosomes were associated with cell-cell adhesion [e.g., E-cadherin (CDH1) and Epithelial cell adhesion molecule (EPCAM)], while cytoskeleton proteins (e.g., Vimentin) and proteases (e.g., MMPs 1/14/19) were upregulated in these EVs [89]. Interestingly, because MMP-14 and -19 were not previously observed in the classical secretome of MDCK/21D1 cells [85], it was suggested that exosomes may represent a specific mechanism to transport these proteases to distant sites [89]. Also using MDCK cells, Gopal and colleagues established a YBX1-overexpressing clone (MDCK^YBX1^ cells) showing a pEMT state [35,90]. Compared with the parental cell line, MDCK^YBX1^-exosomes were shown to contain enriched oncogenic proteins (e.g., KRAS, HRAS, RhoC), and proteins with a known role in angiogenesis [e.g., Integrin subunit beta 1 binding protein 1 (ITGB1BP1)] and cell motility (e.g., Rac1) [35,90]. Additionally, MDCK^YBX1^-exosomes effectively transferred Rac1 to 2F-2B endothelial cells, inducing 2F-2B cell migration in vitro [35,90]. These results are of great relevance by showing that cells undergoing EMT can effectively use these vesicles to transfer cargo to other cell types and induce a behavior largely associated with cancer growth and progression, such as angiogenesis.

Although limited by available techniques, in vitro studies have performed a deep evaluation of the proteome of cancer cells undergoing EMT and their secretome. It is clear so far that different EMT inducers and regulators can trigger similar mechanisms, thus driving cells to a common outcome. However, further analyses are still needed to better characterize the precise extent of the crosstalk and overlap between these mechanisms. Improvements in existing methodologies will allow not only the identification of poorly expressed relevant proteins, but also determine post-translational modifications that may be critical for their stability, localization, and activity. Still, from the understanding obtained from cell culture-based studies, many EMT effectors have been revealed. As discussed here, high-resolution methods have begun to be used to interrogate changes at the sub-cellular level. Similarly, as seen for enriched fractions from cell lysates, much information has also been described by exploring the localization of proteins in EVs [91,92,93,94,95,96]. In this context, the preferential expression of proteins in the membrane or lumen of EVs could help to explain the targeting, interaction, and intracellular fate of EVs in recipient cells. Likewise, determining the source of such proteins carried by EVs may further elucidate the mechanisms of activation for specific signaling pathways and their propensity to regulate EMT in targeted cells. Evaluating the levels, localization, and orientation of such proteins in clinical samples may lead to new targeted therapies and biomarkers with enhanced potential to detect and treat early-stage metastases, as well as to predict cancer recurrence, metastatic progression, and resistance to therapy.

### 3.2. Looking towards New EMT Biomarkers in Primary Tumors by Using Proteomics

Although in vitro models are useful for exploring specific mechanisms that drive or block the EMT process, they usually lack natural intratumor and intertumor heterogeneity otherwise observed in real cancers. In addition, representing the interaction between cancer cells and non-cancer cells within the tumor microenvironment is a complex task. Therefore, characterizing the proteome of only one cell type may also mislead the interpretation of its real significance. In this context, the best approach to understanding the contribution of the EMT to the progression of real tumors may demand a careful analysis of the proteome of tumor samples. Still, as confounding factors might be more difficult to isolate in this scenario than in vitro, additional considerations should be kept in mind, such as the existence of distinctive molecular subtypes and clinicopathological characteristics for the patients included in the study.

Considering that most proteomic techniques are very expensive and time-consuming, studies aiming to evaluate the proteome of human cancers would hardly be designed to exclusively evaluate EMT-related effectors or biomarkers. Otherwise, such proteins may eventually emerge among the set of differentially expressed molecules, particularly considering their well-established relevance in cancer invasion and metastasis, as discussed before. For instance, Moreira et al. (2004) have observed this pattern of differentially expressed proteins when comparing the proteome of bladder specimens derived from normal tissues and transitional cell carcinomas (TCCs) [97]. Sixty percent of the tumors analyzed expressed high levels of Vimentin and PGP9.5 [also known as Ubiquitin C-terminal hydrolase L1 (UCHL1)], suggesting the presence of cancer cells undergoing EMT [97]. Additionally, invasive tumors showed lower levels of the epithelial protein 14-3-3σ (also known as Stratifin) than normal tissues and non-invasive tumors [97]. Interestingly, the evaluation of tumors exhibiting heterogeneous staining for 14-3-3σ demonstrated the progressive loss of its expression, with noteworthy negative staining in invasive areas [97]. Similarly, Sun and colleagues reported that the mesenchymal marker Vimentin was consistently overexpressed in hepatocellular carcinomas (HCCs) compared with cirrhotic and normal liver tissues, reinforcing the association between EMT and cancer progression [98].

The focus on proteins with a known significance to EMT might however mask the detection of new biomarkers contributing to EMT-related effects. Because EMT is frequently associated with cancer progression, an alternative is to group the samples according to the TNM stage, thus considering tumor size, impacts on nearby lymph nodes and metastatic occurrence. Celis and collaborators used this experimental design and reported that the downregulation of keratin-13, alpha-Fatty acid binding protein (α-FABP), Glutathione S-transferase (GST)-µ, and D-3-phosphoglycerate dehydrogenase (PDGH) was correlated with increasing grade in TCCs [99,100]. In BCs, a molecular signature including several EMT markers (e.g., CK5/6, CK8/18, E-cadherin, and P-cadherin) was associated with metastatic progression and successfully used to group cancer patients in poor- or good-prognosis clusters [101]. Also studying the proteome of BCs, Cawthorn and colleagues described 477 proteins differentially expressed between cancer samples from patients affected or not by lymph node (LN) metastasis [102]. Among these proteins, Decorin levels in cancer cells were correlated with LN metastasis in terms of both occurrence and number of LNs affected, whereas the expression of Heat shock protein (HSP)-90B1 was associated with distant metastasis [102]. In bladder cancers, high grade tumors were enriched in mesenchymal markers (e.g., Vimentin, Collagen α-1I), basal and cancer stem cell markers [e.g., Aldehyde dehydrogenase 1 family member A1 (ALDH1A1) and Bcl-2-associated X protein (BAX)], proteins associated with cell migration (e.g., Fibronectin 1 and Vitronectin), and EMT inducers (e.g., TGF-β1) [103].

As discussed before for in vitro studies, the presence of highly expressed proteins can mask the presence of less abundant molecules able to play critical roles in the biological process evaluated. Although sub-proteomes of human tumors have not been frequently evaluated, some examples highlight their relevance. The analysis of the cytosolic fraction of BCs, for example, associated higher levels of Ferritin light chain (FTL) with reduced metastasis-free survival [104,105]. Interestingly, histological analyses revealed that FTL was mostly expressed by stromal cells and its levels were correlated with the expression of CD138 (also known as Syndecan) [104]. Because CD138 is a typical mesenchymal marker, its expression in both stromal cells and cancer cells suggested the occurrence of EMT in at least part of these breast cancers [104].

Altogether, these studies confirm that many EMT-related proteins identified in vitro show potential use as biomarkers for different types of cancer, being associated with cancer recurrence, lymph node metastasis, and distant metastasis. Moreover, while important EMT-related alterations have been characterized in vitro, many proteins are regulated by the interactions between cancer cells and stromal cells. This observation reinforces the importance of integrative studies analyzing the proteome of immortalized cancer cells and human cancer samples. Still, because analyzing proteins from tumor samples demands highly invasive approaches to obtain biopsies and resected samples, this may be a problem when monitoring cancer patients before/after treatment. In the next section, studies focused on overcoming this limitation by investigating and establishing biomarkers in biological fluids are discussed.

### 3.3. Biological Fluids: An Easier Access to EMT-Related Biomarkers

The analysis of tumor samples, particularly their proteomes, requires invasive procedures to obtain enough tissue to detect low-abundance proteins. Otherwise, the evaluation of biomarkers in biological fluids such as blood, saliva, and urine requires less invasive methodologies that enable a closer and easier follow-up of the patients. Moreover, analysis of patient-derived data obtained over time (e.g., before, during, and after therapeutic intervention) may help to more accurately characterize the dynamic networks that regulate such a fluid process as the EMT.

Aiming to establish biomarkers for bladder cancers, research led by Celis [106] and Ostergaard [107] reported increased levels of Psoriasin (also known as S100A7) in squamous cell carcinomas (SCCs) and urine samples from cancer patients. Interestingly, Psoriasin was not observed among the serum proteins of SCC patients, indicating its specific use as a biomarker to be screened in the urine of these patients [106,107]. In another example, Sun and colleagues reported Vimentin as a sensitive and specific biomarker when analyzing the proteome of serum samples from HCC patients and non-neoplastic controls [98]. In this study, Vimentin levels were used to distinguish even patients with small HCCs (<2 cm) from non-neoplastic controls [98].

Whereas the detection of cancer proteins in biological fluids shows clear benefits, it also incurs a methodological issue associated with the presence of highly expressed proteins that may mask less abundant molecules. To improve the detection of relevant molecules, particularly from serum and plasma samples, the depletion of excessively abundant proteins is recommended. In BC patients, proteomic analysis of albumin-depleted serum samples demonstrated the association between LN metastasis and several proteins related to the cytoskeleton and ECM structure or remodeling, including collagen α4I, Serpin C1, Fibrinogen gamma chain (FGG), and Tenascin XB (TNXB) [108]. Moreover, in this study, TNXB was detected in the serum samples of all patients diagnosed with benign breast diseases and LN-negative cancer patients, but not in LN-positive patients, indicating that loss of circulating TNXB could be used as a biomarker of LN metastasis [108].

Several cancer types lack specific diagnostic or prognostic biomarkers. Even for cancers better characterized (e.g., breast, colorectal, and lung cancers), only a small number of biomarkers exist, and their use is restricted by reduced sensitivity and specificity. Moreover, many different conditions with clinical relevance cannot be determined by current biomarkers, including the progression toward resistance to anti-cancer therapies and the development of locoregional and distant recurrence. As observed by the studies discussed here, EMT-related proteins could be used as cancer biomarkers, particularly if considering their typical association with cancer progression and metastasis. Still, the reduced number of publications evaluating cohorts specifically grouped according to EMT-associated outcomes limits the generalization of the conclusions obtained. For instance, additional proteome-based studies including patients who progressed to LN metastasis or distant metastasis, may confirm the reliability of EMT-related proteins for this purpose. Investigations focused on recurrence and resistance to therapy have also been overlooked, and cancer types with lower incidence have been often ignored in this kind of evaluation. In addition to an experimental design focused on EMT-associated modifications, improved technologies may allow the characterization of cancer protein profiles, as has been currently performed for the characterization of cancer transcriptomes.

In the next section, new perspectives to integrate multiomics data from real tumors are discussed to highlight how these approaches can further increase our understanding of EMT. Also, we prospect a better comprehension of the proteome of EVs and CTCs led by emerging methods with superior sensitivity. In addition to EVs and CTCs, improved methods may enable characterizing the proteome of cancer cells and non-cancer cells in the tumor microenvironment from a spatio-temporal point of view.

## 4. Integration of Multiomics and Spatio-Temporal Analyses for a Comprehensive Understanding of EMT-Driven Cancer Progression

In contrast to studying the changes at the DNA/RNA level, current proteomic methods cannot satisfactorily cover the entire diversity of proteins within cancer cells or the tumor mass. Additionally, evaluating the protein profile of multiple samples is highly expensive and time-consuming. These and other reasons led to a better understanding of how modifications in the epigenome and transcriptome impact biological processes while depicting global changes in protein levels was left behind. But to what extent should we rely on a single omics when translating EMT-related findings to the clinic? Although several studies individually demonstrated global alterations either in the epigenome, the transcriptome, or the proteome of cells undergoing EMT, the direct comparison of these results is still an issue. Besides analyzing different cancer types, in vitro studies frequently induce EMT by exploring only one EMT inducer at a time. In addition to differences across studies, this strategy limits the investigation of possible crosstalk between multiple signaling pathways. Therefore, multiomics may provide a more reliable source for the comparison of the many regulatory mechanisms impacting cancer cells at multiple levels during EMT—particularly when analyzing cancer patient samples that are inherently affected by several EMT regulators.

In fact, seminal studies published throughout the last decade have already begun to adopt multiomics as an approach to analyze alterations in cancer samples that simultaneously impact DNA, RNA, and protein levels. Although most of these studies investigated different cancer types, some similarities are worth mentioning. Among them is the common divergence between transcriptome- or proteome-derived data, such as that reported in colorectal [109], breast [110,111], ovarian [112], gastric [113], and lung [114] cancers. Besides methodological parameters influencing this correlation, spatio-temporal changes in RNA species largely affect their localization and availability for translation, thus, impacting protein levels [115,116,117,118]. Importantly, this effect is reported to change in pathological conditions, being increased in cancers when compared with normal tissues, and particularly enhanced with cancer progression [115]. Moreover, copy number alterations (CNAs) and post-translational modifications are also shown to significantly impact gene expression in a way that is not necessarily translated into protein modifications [110,114]. In addition to reinforcing an important difference in the mechanisms that impact cancers at the molecular level, this analytical divergence has a profound impact on the ability to determine patient prognosis. Remarkably, depending on the method of choice, patients showing significantly different probabilities of survival cannot be distinguished by such omic analysis. For example, in CRCs, only a proteome-based clustering—but not other types of analysis—revealed a typical EMT signature correlated with poor prognosis [109]. Furthermore, the identification of a molecular subtype associated with cell invasion and poor survival rate in early-onset gastric cancers (EOGCs) required an integrative clustering using global mRNA, proteome, phosphoproteome, and N-glycoproteome [113]. This observation reinforces a critical problem as the use of single omic methods may not suffice to accurately describe the myriad of alterations within a tumor, therefore, representing an obvious issue in determining therapeutic approaches.

Although the integration of multiple omics helps to overcome limitations otherwise imposed by the individual use of each approach, spatial alterations are often masked in bulk analysis, and detecting temporal modifications remains unfeasible—especially considering a clinical context. As mentioned before, the increasing association of deconvolution strategies and transcriptomic-focused methods with single-cell resolution has been instrumental in depicting the contribution of different tumor compartments regarding EMT-related alterations. Tissue microdissection also partially helps to overcome this limitation to individually investigate molecular signatures associated with either tumor epithelium or stroma. For instance, in microdissected prostate cancers, a gradual decline in phosphorylated (p-) Mitogen-activated protein kinase (ERK) levels and concurrent increase in p-AKT levels have been associated with cancer progression [119]. Similar results were also reported in microdissected CRCs, where decreased p-ERK and p-p38 levels were observed in cancer tissues compared with uninvolved mucosa [120]. Thus, integrating tissue microdissection and tissue microarray may increase the understanding of spatial modifications otherwise overlooked by the analysis of bulk samples.

Further, a combined approach can also help to characterize alterations in rare samples that are not located within the cancer mass but have been shed by the tumor and may be scattered throughout the body, such as CTCs. For instance, in a study simulating CTCs by spiking immortalized cancer cells into blood samples, 4000 proteins were identified by one-dimensional high-resolution porous layer open tube-liquid chromatography (LC)-MS in samples spiked with 100–200 MCF7 breast cancer cells [121]. Impressive results were also described by using nano-LC-MS for the analysis of 1–5 LNCaP prostate cancer cells spiked and recovered from blood samples [122]. In HNSCC patient samples, the use of mass cytometry and unsupervised clustering allowed the identification of epithelial and EMT sub-groups of CTCs, where the latter accounted for more than 80% of all CTCs [123]. Interestingly, the expression of immune checkpoint proteins (e.g., PD-L1 and CTLA4) was lower in CTCs with an EMT phenotype when compared with epithelial counterparts [123]. Further, analysis of molecular pathway activity revealed that CTCs expressing EMT traits were also enriched in p-CREB and p-ERK proteins, but showed reduced levels of other intracellular effectors, such as p-STAT3, p-STAT5, p-PARP, and p-AKT [123]. Establishing the profile and significance of immune checkpoints and intracellular effectors in CTCs may have a profound impact on the development of therapeutic strategies focused on overcoming metastatic progression and resistance to therapy. Noteworthy, while methodological improvement is still required to analyze the proteome of patient CTCs, innovative studies have already begun to characterize the genome, transcriptome, and metabolome of these shedded cells [124,125,126,127,128,129,130,131,132,133,134]. Such analyses have not only helped to elucidate how mutations and molecular programs impact the dissemination of cancer cells but have also validated the perspective of employing a multiomic strategy to improve our knowledge of cancer progression.

As for CTCs, few studies have comprehensively investigated the protein profile of EVs isolated from cancer patient biofluids. Nevertheless, initial studies in circulating EVs have already demonstrated an association between HCC progression and increased Galectin-3-binding protein (LG3BP) levels [135]. In CRC, Transferrin receptor protein 1 (TFR1) was reported to be enriched in circulating EVs from non-metastatic patients [136]. In BC, the establishment of protein signatures for circulating EVs (including EGFR, p-cadherin, and fibronectin) enabled differentiating cancer patients from healthy subjects and was further associated with cancer progression, and relapse [137]. Moreover, while the isolation and characterization of EVs from biofluids remains challenging, the development of microfluidic devices for the isolation and enrichment of such membranous particles brings interesting possibilities for the diagnosis and monitoring of cancer patients. For example, it has been recently reported that the analysis of epithelial and mesenchymal markers on plasma EVs captured by microfluidic devices can be successfully used to establish the prognosis of patients with pancreatic cystic lesions [138]. This strategy is particularly important as it uses tumor-derived EVs to monitor EMT dynamics in pancreatic cells and further inform on whether these patients may or may not undergo surgery [138]. Similarly, quantification of the EMT markers in melanoma-EVs through microfluidic devices has been described as an innovative strategy to monitor disease progression. In this context, increased levels of mesenchymal markers (N-cadherin and ABCB5) compared to epithelial markers (E-cadherin and THBS1) characterized a shift in the serum EVs of melanoma patients that was also correlated with the development of drug resistance [139]. Overall, although limited in number, the significance of these studies is remarkable and may be increased if combined with those where DNA and RNA species transported by cancer patient EVs are analyzed and also associated with diagnostic or prognostic potential [140,141,142]. Furthermore, since EVs and CTCs can both be obtained from blood samples and separated based on physicochemical properties, new methods are emerging to optimize their sequential isolation and analysis in a parallelized multidimensional analytic framework [143,144]. Importantly, such methods must not be understood simplistically as additional strategies for the discovery of EMT-related biomarkers. Rather, innovations improving the analysis of rare samples in liquid biopsies (e.g., CTCs and EVs) are paramount to generate a holistic view of the signaling pathways underlying EMT while also providing information on its dynamic regulation during metastasis. In this scenario, biomarkers emerge from an in-depth understanding of the molecular machinery that drives disease progression. Consequently, the translation of such biomarkers into clinical practice will improve existing diagnostic and monitoring methods due to enhanced specificity and sensitivity.

Altogether, the results discussed here demonstrate the possibility of evaluating the proteogenome of small samples, but future studies are still needed before implementing such analyses in clinical routine. While in vitro experiments benefit from “unlimited” starting material, human samples are finite, restricting the number of tests and especially compromising the detection of rare proteins. In addition to increasing the sensitivity of proteomics and their integration with epigenomics/transcriptomics, current studies highlight the importance of improving methods for the isolation of scarce samples, such as EVs and CTCs that are naturally obtained at small concentrations. Overcoming these limitations may lead to novel mechanisms and biomarkers to be used for the diagnosis and treatment of cancer patients (Figure 4).

## 5. Future Directions

Although targeted therapies against some of the major EMT-inducers have been vastly used to treat cancer patients (e.g., anti-EGFR tyrosine kinase inhibitors), others have repeatedly failed in clinical trials (e.g., anti-TGF-β inhibitors) [15,145]. Consistent with this notion is the delayed development of efficient therapies against metastasis and recurrence, while major achievements have been seen over the years for drugs targeting primary tumor growth [146]. This observation highlights major gaps in our understanding of the EMT program, which consequently impair the development and the adequate use of anti-EMT therapies in the treatment of cancer patients. As discussed throughout this review, addressing this issue will likely involve the integration of multiomics and spatio-temporal approaches to precisely characterize the crosstalk between the multiple molecular pathways that regulate EMT and MET during cancer progression. Ideally, new technologies and methods will improve our understanding of the architecture of the tumor microenvironment while simultaneously allowing the investigation of alterations taking place in intracellular and extracellular compartments. For instance, structures such as endosomes, lysosomes and extracellular vesicles are highly specialized, and the trafficking of different molecules along them is tightly regulated. Precisely characterizing the spatio-temporal alterations impacting molecules that are trafficked through these compartments and used in cell-cell communication might inform about the fate of cancer cells and disease progression.

Moreover, whereas important signaling pathways able to induce EMT-related cancer progression have been described, the existing methods used for the diagnosis and monitoring of cancer patients have rarely taken advantage of such well-defined knowledge. More specifically, the poorly efficient methods currently used in clinics still fail to efficiently distinguish healthy subjects and different cancer stages because of intrinsic conceptual limitations that overlook the impact of EMT inducers and regulators. Underlying such issues is the common adoption of biomarkers that are not involved in the core molecular program that ultimately promotes cancer progression. Accordingly, these methods are typically associated with reduced sensitivity and specificity. Thus, overcoming these limitations to improve patient diagnosis and monitoring will require the establishment of sensitive methods with a focus on detecting and tracking alterations in biomarkers with real impact on cancer progression, especially EMT drivers [147,148,149].

Furthermore, it is worth noting that the clinical samples analyzed in many studies (as those discussed here) are usually collected before treatment or, at best, once following the administration of anti-cancer therapies. Such a strategy may mask several alterations with critical relevance for the progression of the disease that cannot be detected when patient samples are collected and analyzed too long before or after treatment. This may be particularly important if considering low-abundance biomarkers that are unique to a small fraction of cells (or EVs) with elevated metastatic potential. Thus, a closer monitoring of cancer patients and the analysis of multiple samples collected over time might lead to the identification of alterations in molecules associated with response to treatment, therefore, helping to guide patient care. As omic-based methods may not be appropriate in this context, they may be used in conjunction with other methods of cheaper and quicker execution (e.g., ELISA and qPCR). Moreover, as sampling fragments of the tumor may not be suitable for repeated analyses, using biological fluids would be advantageous as they require less invasive techniques to be collected, thus preserving the quality of life of cancer patients.

## 6. Conclusions

Whereas the past decades have been a stage for many studies focused on characterizing EMT drivers and regulators, efficient diagnostics and therapies based on these molecules are still missing. This highlights a major problem in precisely characterizing the mechanisms that promote cancer progression and our failure to use well-established mechanisms to benefit cancer patients. Thus, an innovative strategy to overcome such limitations is urgently needed.

EMT is an intricate process that involves the crosstalk between multiple signaling pathways [17,150,151,152,153,154,155]. Many signaling pathways controlling cancer cell EMT are also required for tissue homeostasis, which explains their tight regulation. Characterizing the mechanisms controlling EMT and the myriad of associated molecules is an arduous task that favors overlooking those molecules that are poorly expressed, irrespective of their potential significance in clinics. Although interrogating changes in one molecular pathway at a time has been crucial to depicting the EMT program, integrating multiple omics may help reveal the persisting blind spots that still compromise our understanding of this process. While changes at DNA and RNA levels reflect quick responses exhibited by cancer cells and non-cancer cells, alterations in protein levels and post-translational modifications impact their localization, stability, and activity. Similarly, while investigating modifications in the genome and transcriptome of these cells is favored by high sensitivity due to amplification-based methodologies, alterations in their proteome may more accurately reveal the real drivers of EMT-related phenotypic changes.

Furthermore, besides combining multiple omic techniques, improvements in current in vitro and in vivo methods will increase our understanding of the real complexity of EMT-related alterations. At the same time, the integration of pre-clinical studies with improved real-world data will lead to better options in the fight against cancer metastasis and recurrence. In the context of diagnostics and treatment, this will enable obtaining more precise and useful information about EMT-related changes to monitor disease progression in (nearly) real-time to ultimately improve patient outcomes.

## Figures and Tables

**Figure 1 cells-12-02740-f001:**
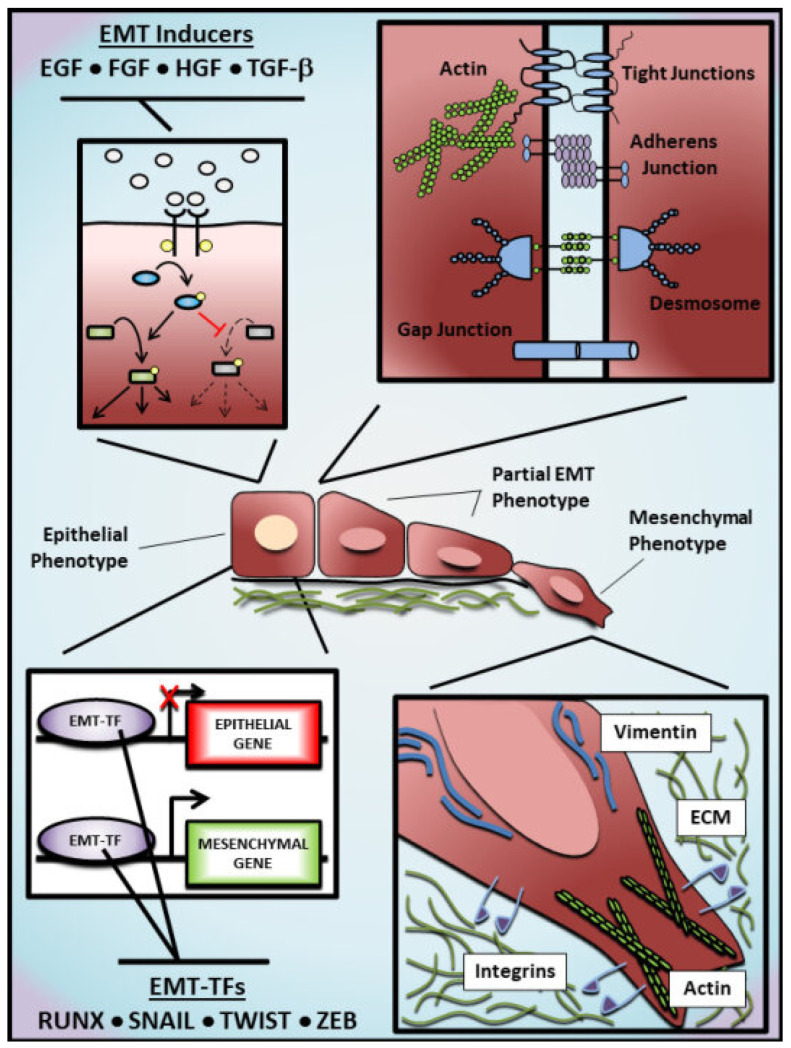
Molecular regulation of epithelial-mesenchymal transition. The epithelial-mesenchymal transition (EMT) represents a continuum of alterations from an epithelial phenotype towards a mesenchymal phenotype, passing through diverse intermediate EMT states. Growth factors such as Epithelial growth factor (EGF), Fibroblast growth factor (FGF), Hepatocyte growth factor (HGF), and Transforming growth factor-beta (TGF-β) are frequently used as EMT inducers in experimental models to activate/inhibit downstream effectors and trigger an EMT program that includes alterations in the epigenome, transcriptome, and proteome of stimulated cells. Among these alterations, EMT transcription factors (EMT-TFs), such as RUNX, SNAIL, TWIST and ZEB family members, are positively regulated to inhibit the transcription of epithelial genes (e.g., *CDH1*, *OCLN* and *TJP1*) and promote the transcription of mesenchymal genes (e.g., *CDH2*, *VIM* and *ACTA2*). As an alternative to EMT inducers, EMT can also be experimentally promoted by the overexpression of EMT-TFs. Cells undergoing EMT exhibit the dissolution of protein complexes responsible for cell-cell adhesion, losing the apicobasal polarity and acquiring a back-front polarity that is characterized by the presence of actin stress fibers and an elongated morphology. In addition, these cells show increased migratory and invasive potential, which is commonly associated with basement membrane degradation and extracellular-matrix remodeling. Arrows shown in the top left represent the activation (black) or inhibition (red) of hypothetical signaling pathways by EMT inducers. Dashed arrows represent the attenuation of the activity of hypothetical effectors.

**Figure 2 cells-12-02740-f002:**
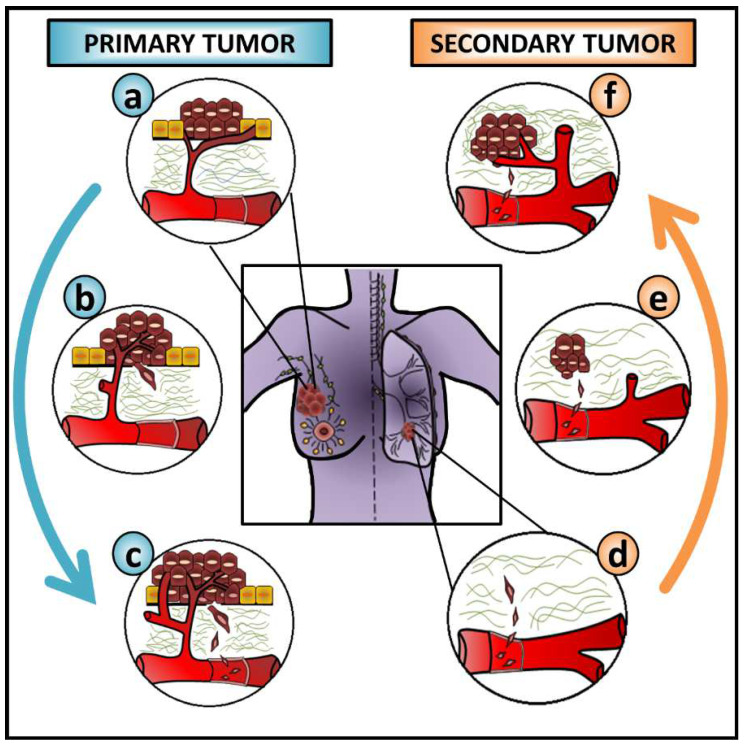
The role of EMT and MET during cancer metastasis. This figure exemplifies the contribution of epithelial-mesenchymal transition (EMT) and mesenchymal-epithelial transition (MET) during cancer progression when breast cancer cells metastasize to the lung. Overall, this is expected to be common to many carcinoma cells from different cancer types metastasizing to distinct sites. Cancer cells from (**a**) the primary tumor (**b**) undergo EMT, degrade the basement membrane and the ECM, and invade surrounding tissues before (**c**) blood vessel intravasation. After (**d**) the extravasation of circulating tumor cells (CTCs) into a distant organ, their interaction with the local microenvironment stimulates cancer cells to undergo (**e**) MET, seeding and colonizing a secondary site, and allowing (**f**) the growth of a secondary tumor. The blue arrow on the left side of the figure and the orange arrow on the right side of the figure represent the EMT and MET progress, respectively.

**Figure 3 cells-12-02740-f003:**
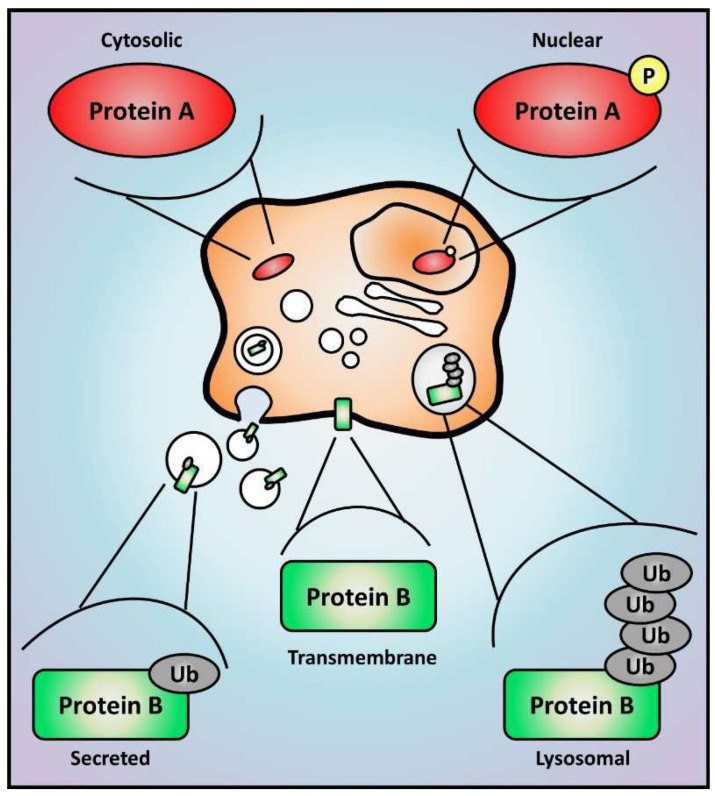
Proteomic analysis determines protein expression, activation, and localization. The study of cell proteome offers an unbiased view of global protein levels, and enrichment methods can also reveal major alterations in post-translational modifications that are able to impact protein activity, stability, and localization. Hypothetical changes are represented here for proteins A and B. Protein A translocates from the cytoplasm to the nucleus if phosphorylated, which could represent its activation as a transcription factor. Protein B is a plasma membrane protein that can be secreted into extracellular vesicles if monoubiquitinated or degraded in lysosomes if polyubiquitinated. Yellow circles and grey ellipses represent protein phosphorylation (P) and ubiquitination (Ub), respectively.

**Figure 4 cells-12-02740-f004:**
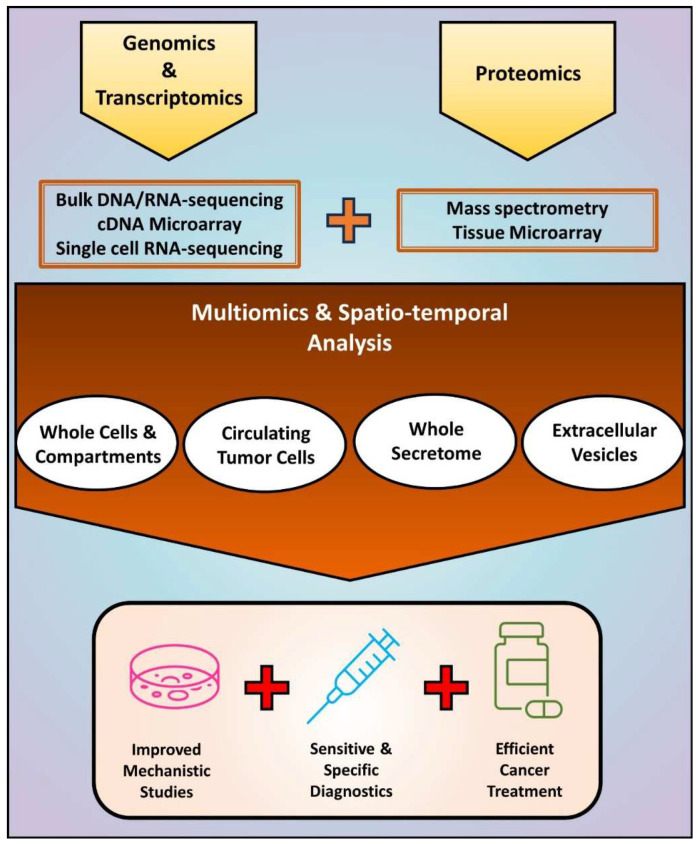
An integrated approach based on multiomics and spatio-temporal analysis of EMT-driven cancer progression. Current studies evaluating EMT at DNA, RNA, and protein levels have relied on the independent use of methods such as bulk DNA/RNA-sequencing, cDNA microarray, single cell RNA-sequencing, mass spectrometry, and tissue microarray. The combination of these methodologies enables obtaining information about simultaneous alterations in cancer cells and fractions that include the intracellular and the extracellular milieu. In addition to improving mechanistic studies in vitro and in vivo, such strategy may lead to more sensitive and specific diagnostic methods, and efficient anti-cancer therapies able to prevent and treat EMT-driven metastasis and recurrence.

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
