# Peer review of "A New Era of Integration between Multiomics and Spatio-Temporal Analysis for the Translation of EMT towards Clinical Applications in Cancer"

_cells, 2023, doi:10.3390/cells12232740_

Round 1

Reviewer 1 Report

Comments and Suggestions for Authors

The authors have provided a commendable review on the necessity of multiomics data in enhancing our understanding of the role of EMT in cancer studies. They astutely recognize that an integrative multiomics approach is paramount in dissecting the complexities of EMT, with an emphasis on how proteomics can contribute to this field. Their work underscores the potential that lies in harnessing the power of combined omics strategies to unravel the molecular intricacies of EMT-driven metastasis.

Just one comment which the author did not clear described. It is crucial to consider the potential confounding factor of stromal cell contamination in the multiomics datasets, particularly those derived from patient samples. This contamination can significantly skew the data interpretation, leading to inaccurate conclusions. Therefore, employing single-cell and spatial transcriptomics analyses can be instrumental in distinguishing the signals from different cell populations within the tumor microenvironment, ensuring a more accurate attribution of molecular changes to specific cell types involved in EMT.

 Furthermore, temporal sampling, such as the analysis of circulating tumor cells (CTCs), could offer valuable insights with enhanced sensitivity, allowing for the monitoring of EMT dynamics and its contributions to tumor progression over time. Such longitudinal data can provide a more detailed and dynamic picture of EMT, offering clues to its transient and reversible nature in the context of metastasis.

 The pursuit of multiomics data in the study of EMT is indeed invaluable. Not only does it offer a holistic view of the underlying mechanisms, but it also holds the promise of identifying key drivers that could be leveraged for therapeutic advances. By integrating and interpreting multiomics data, we can begin to elucidate the regulatory networks steering EMT and potentially translate these findings into more effective cancer treatments. The authors present a nice review about these.

Author Response

Authors agree and appreciate the reviewer’s comments. In the original version of this manuscript, we have briefly mentioned that non-cancer cells may negatively impact the use of EMT signatures in bulk transcriptomic analyses. Following the reviewer’s suggestion, this point has been expanded to add clarity and emphasis on its relevance. Modifications in this regard are highlighted in the updated version of this manuscript (lines 251-254, lines 260-272, and lines 608-610). Similarly, whereas the initial version of this manuscript has discussed the evaluation of EMT-related alterations in CTCs and EVs, this has been expanded by incorporating additional references aiming at emphasizing its significance for temporal analysis, specially in respect to monitoring disease progression. Changes focused on the analysis of CTCs and EVs are also highlighted in the updated version of this manuscript (lines 519-251, lines 626-635, lines 649-661, and lines 664-675).

Reviewer 2 Report

Comments and Suggestions for Authors

The review article submitted by Adilson Fonseca Teixeira et al focuses on Epithelial-mesenchymal transition (EMT) as important player in cancer research. In particular, the authors discuss evidence of several gaps in the understanding of the EMT program and how those gaps affect the development of anti-EMT therapies in the treatment of cancer patients. Also, the authors strongest suggest that “the integration of multiomics and spatio-temporal approaches are need to precisely characterize the crosstalk between the multiple molecular pathways that regulate EMT and MET during cancer progression”. The text and data are presented in a rational way, making it easy to read and understand. The idea proposed in the abstract of this review article is original and exciting. I’m very enthusiastic about this manuscript and I have no major concerns.

Author Response

We welcome the kind comments from reviewer #2 and their appreciation about our work conducted during the writing of this review manuscript.

Reviewer 3 Report

Comments and Suggestions for Authors

In this manuscript, the authors have attempted to describe the use of spatiotemporal and multi-omic analyses to investigate the phenotypic transition during cancer progression.

Overall, the manuscript covers the topic in a not-very-innovative way. The first three chapters describe the EMT phenomenon; such characterisations can be found in many other publications. Moreover, these chapters lack references from the last 2-3 years, and the list of references is dominated by those from the previous 5-10 years.

Only the fourth chapter is related to the title of the manuscript. However, the information presented there is superficial and not systematised, presented randomly, in my opinion.

The manuscript's content is generally unrelated to the title, and the part suggested by the title is superficial and incoherent; the authors have not attempted an in-depth analysis of the problem.

Author Response

After carefully discussing the reviewer’s comments about this manuscript, we would like to express our appreciation for their considerations, particularly as they intended to improve the quality our work by the highlighting the need for innovative perspectives and in-depth analysis of the literature. However, we disagree with the assessment on the overall qualification of this manuscript quality and coherence.

In respect to the three first sections of this manuscript, it was our intention to start by introducing the main topic of this review article (‘section 1. Introduction to EMT’). It seems reasonable to include an introductory section in review articles and we must assume most (if not all) reviews follow a similar structure in this sense. Next, we aimed at contextualizing the main strategies that have been used so far to discover and characterize the molecular changes observed in cancer cells undergoing EMT (‘section 2. Benefits and pitfalls of analyzing cancer cell EMT at DNA/RNA level’ and ‘section 3. Proteomics translated from bench-to-bedside’). Although we agree that part of the content presented in these sections may be found in other review articles, we understand that such context must be presented before any perspectives are introduced and discussed. Moreover, whereas we understand that such sections might not be particularly new for experts in the field, they were specifically planned to welcome researchers not yet familiarized with the topic. Thus, trying to establish a balance between presenting this content to a new audience without prolonging it excessively, we relied on selected references to exemplify the limitations that are imposed to single omics analyses. Noteworthy, much of the early research done on the transcriptome and proteome of cancer cells undergoing EMT was not published in the last two years, but rather published over 5 years ago. This observation explains our decision to include these studies in our review article. Accordingly, it is also reasonable to assume that  by acknowledging the contribution of pioneers in the EMT field, a proportionally high number of so-called ‘old’ original articles would be included in list of references. Yet, we also understand and agree with the need for highlighting the current status of the field. Thus, we have updated the list of references accordingly. Modifications are highlighted throughout the revised manuscript (lines 35, 37, 70, 126, 127, 170-172, 189-200).

In regard to the fourth section of this manuscript, our intention was once again to present some of the early research done in the field through selected references where multiomics were used to investigate cancer patient samples. We did not consider fruitful to describe each these early investigations in detail. Rather, we have specifically planned to discuss what have prevented the successful translation of EMT-related findings into significant benefits for the diagnosis, treatment, and monitoring of cancer patients so far. Limitations associated with the translation of EMT-related findings cannot be exclusively attributed to problems with the classification of epithelial, mesenchymal, or intermediate states in cancer cells from patient samples. Rather, the unsuccessful translation of EMT into clinics is highly associated with the transient and reversible nature of this process. In other words, the problem lies on accurately investigating the role of EMT on cancer progression while considering all the spatial and temporal changes observed in human cancers. In this context, we disagree the reviewer’s comment concerning the superficiality or lack of systematization of this manuscript. After initial sections describing the spatial and temporal limitations of using single omics, we have introduced and discussed possibilities to overcome such limitations paragraph-by-paragraph. Specifically, since depicting EMT-related effects remains a problem when using bulk omics, we discuss strategies that help to overcome spatial limitations such as by integrating multiple omics, scRNA-seq and tissue microdissection. Also, since tumor sampling over time is not feasible, we present and discuss evolving methods to isolate, characterize and analyse rare samples (i.e., CTCs and EVs), which also include using less invasive methods (e.g., liquid biopsies). Noteworthy, while many recent articles have presented interesting uses for such strategies, a gold standard is still missing. Furthermore, very few of these articles focus on EMT or minimally investigate the occurrence of this process in cancer patient samples. As discussed in the section 5 (‘Future directions’), this problem is highlighted by the lack of efficient anti-EMT therapies or EMT-related diagnostics in clinics.

Round 2

Reviewer 3 Report

Comments and Suggestions for Authors

I thank the authors for their comprehensive discussion. From their responses, I conclude that they generally agree with my opinion that the first 3 parts of the article, which constitute the largest part of the manuscript, describe the phenomenon of EMT, and only one chapter deals with multi-omic analyses. If we agree on this point, I suggest that the title of the article is changed to a more relevant one that reflects the majority of the content and that multi-omic analyses be removed from the title and focus only on the mechanisms of the EMT process, which will be more useful for readers looking for specific information. It is regretted that the authors did not decide to present the content of Chapter 5 in the form of tables or summaries, which would have been much more user-friendly for the readers.

Author Response

The authors welcome reviewer's reversed and improved scores. The reviewer's comments, we believe, are based on the presumption that review needs to be written in a particular way. Indeed most reviews are written in the way the reviewer prefers. In fact there are more than enough reviews being written in such way on the topic of EMT and omics. However the authors aim and focus of the manuscript are different here, ie identifying research gaps and pointing to real needs towards meaningful clinical advances. As such we are not persuaded by merit of argument to change the title of manuscript.